# Epicardial Adipose Tissue: A Novel Potential Imaging Marker of Comorbidities Caused by Chronic Inflammation

**DOI:** 10.3390/nu14142926

**Published:** 2022-07-17

**Authors:** Maria Grazia Tarsitano, Carla Pandozzi, Giuseppe Muscogiuri, Sandro Sironi, Arturo Pujia, Andrea Lenzi, Elisa Giannetta

**Affiliations:** 1Department of Medical and Surgical Science, University Magna Grecia, 88100 Catanzaro, Italy; mariagrazia.tarsitano@unicz.it (M.G.T.); pujia@unicz.it (A.P.); 2Department of Experimental Medicine, Sapienza University of Rome, Viale Regina Elena, 324, 00161 Rome, Italy; carla.pandozzi@uniroma1.it (C.P.); andrea.lenzi@uniroma1.it (A.L.); 3School of Medicine and Surgery, University of Milano-Bicocca, 20126 Milan, Italy; g.muscogiuri@gmail.com (G.M.); sandro.sironi@unimib.it (S.S.); 4IRCCS, Istituto Auxologico Italiano, 20126 Milan, Italy; 5Department of Radiology, ASST Papa Giovanni XXIII Hospital, 24127 Bergamo, Italy

**Keywords:** epicardial adipose tissue, inflammation, cardiovascular diseases, diabetes, metabolic syndrome, obesity, cardiometabolic risk, echocardiography, coronary CT, CMR

## Abstract

The observation of correlations between obesity and chronic metabolic and cardiovascular diseases has led to the emergence of strong interests in “adipocyte biology”, in particular in relation to a specific visceral adipose tissue that is the epicardial adipose tissue (EAT) and its pro-inflammatory role. In recent years, different imaging techniques frequently used in daily clinical practice have tried to obtain an EAT quantification. We provide a useful update on comorbidities related to chronic inflammation typical of cardiac adiposity, analyzing how the EAT assessment could impact and provide data on the patient prognosis. We assessed for eligibility 50 papers, with a total of 10,458 patients focusing the review on the evaluation of EAT in two main contexts: cardiovascular and metabolic diseases. Given its peculiar properties and rapid responsiveness, EAT could act as a marker to investigate the basal risk factor and follow-up conditions. In the future, EAT could represent a therapeutic target for new medications. The assessment of EAT should become part of clinical practice to help clinicians to identify patients at greater risk of developing cardiovascular and/or metabolic diseases and to provide information on their clinical and therapeutic outcomes.

## 1. Introduction

In the last few years, evidence has accumulated on the relationship between obesity and a wide range of diseases, such as diabetes, non-alcoholic fatty liver disease (NAFLD), cancer, hypertension, cardiovascular diseases (CVD), and neurodegenerative diseases. This association leads to increased morbidity and mortality. The observation of the correlations between obesity and chronic metabolic pathologies has led to the emergence of strong interests in “adipocyte biology”, up to the definition of the adipose tissue (AT) such as a crucial endocrine tissue able to secrete a wide range of hormones and cytokines, which negatively impact on other organs and systems. Therefore, the traditional view of AT as a passive store of excess calories has evolved to implicate an endocrine role [1]. This endocrine function is the result of a complex interaction between adipocytes and cells of the stromal vascular fraction of AT, which modulate mediators produced in different physiological and pathological conditions. Specifically, this important function seems to be assigned to the white adipose tissue (WAT), one of the two major types of AT. WAT is composed of unilocular adipocytes specialized in the storage of energy and the regulation of metabolic homeostasis by the production of adipokines. The other type of adipose tissue is defined as brown AT (BAT) and includes mitochondria-rich multilocular adipocytes specialized in energy dissipation through thermogenesis [2]. Functions of the AT are modulated by cells of the immune system, both innate and adaptive, capable of expounding inflammatory or anti-inflammatory actions depending on the microenvironment and stimuli they receive [3]. In conditions of non-obesity, the AT is characterized by normotrophic adipocytes, and these cells secrete mostly cytokines with anti-inflammatory properties: IL-4, IL-13, IL-10, adiponectin. This “protective” condition is compromised when the AT expands and becomes “obese” and dysfunctional. The initiation of the inflammatory process would be relegated to the hypoxia that is created during the expansion of the AT, with increased adipocyte hypertrophy, hyperplasia and apoptosis accompanied by alterations in the production of adipokines and other mediators [3,4].

WAT is further divided into two main subgroups with different properties and microenvironments: subcutaneous (scWAT) and visceral WAT (VAT). VAT includes epicardial (EAT), perivascular (PVAT), epidydimal (EpiWAT), mesenteric (MAT) and perirenal (PRAT) AT. In particular, the AT surrounding the heart can be divided into—Epicardial adipose tissue (EAT), located between the myocardium and the visceral layer of the pericardium without an intervening fascial plane. The portion directly encompassing the coronary arteries represents the pericoronary adipose tissue (PCAT). Pericardial adipose tissue (PAT) is situated between the visceral and parietal layers of the pericardium. Paracardial adipose tissue is instead located externally to the parietal pericardium [5]. EAT distribution is more concentrated in the atrioventricular and interventricular heart grooves and, when it increases, can cover the ventricles and the epicardial surfaces. It represents 15% of the cardiac mass and also surrounds the adventitia of coronary arteries. EAT has a cardioprotective role and is able to prevent lipotoxicity and secrete anti-inflammatory and anti-atherogenic adipokines during healthy conditions as well as fatty acids and pro-inflammatory cytokines under metabolic insults [6], as shown in several endocrinological diseases [7,8].

The signaling relationship shared between adipose tissue and vasculature has an important role in metabolic regulation. In the classical “inside-out” theory, atherosclerosis is the result of an injury to intimal endothelial cells with a consequent accumulation of inflammatory cells in the subendothelial space. However, according to the “outside-to-inside” theory, even EAT inflammation may contribute to atherosclerosis through a process that begins in EAT and then propagates inward to the vasculature because adipokines and other fat-derived mediators elicit changes within the vessels [9]. Moreover, this relationship is not simply unidirectional, as emerging evidence suggests that mediators derived from the vessel wall affect distinct changes to adipose tissue on a local basis [10]. In particular, the nominal function of human adipose tissue becomes deranged under adverse cardiometabolic/stressful conditions, such as excessive energy supply, obesity, insulin resistance, and diabetes [11]. These may induce phenotypic changes in the EAT including hypertrophy, failure of triglyceride storage, increased lipolysis and release of free fatty acids, and inflammation, with a shift in the secretome of dysfunctional EAT [12], as simplified in Figure 1.

Dysregulation of the adipokines involved in insulin sensitivity, basal metabolism, immune system functions, regulating food intake, fatty acids oxidation and general homeostasis occurs [13]. In particular, in obesity, resistin and leptin increase enhancing the production of pro-inflammatory cytokines such as TNF-α and IL-6, inducing ROS production and stimulating Th1 and Th17 immune responses. At the same time, obese patients have adiponectin and omentin levels markedly reduced [6]. Adiponectin has an important insulin-sensitizing and cardioprotective role, antioxidant activity and immunomodulatory action on several immune cells [14]. Omentin has similar anti-atherogenic and anti-inflammatory functions [15]. Adiponectin and omentin expression are also decreased in patients with CVD, while leptin, resistin and chemerin increase [6]. In addition, the level of pro-inflammatory adipokine chemerin was positively correlated with the severity of coronary atherosclerosis [16]. Moreover, when comparing EAT and scWAT with immunohistochemistry, it emerged that infiltrating CD3+ T cells, tryptase+ mast cells and CD68+ macrophages, and levels of IL-6, IL-1β, MCP-1 and TNF-α were higher in EAT [6]. To give a more recent and direct example of the importance of an improper activation of cytokine systems, it is enough to mention the most recent events relating to coronavirus disease 2019 (COVID-19). Indeed, in COVID-19, obesity and related comorbidities increase the risk of intensive care unit hospitalization and death, due to a rise in inflammation status, impaired immune response and respiratory dysfunction [17]. Obesity appears to be an independent risk factor in young males, metabolic syndrome to promote the cytokine storm, diabetes mellitus and hyperglycemia to impair the immune response and act to ACE2 glycosylation, favoring the infection and pathogenesis of severe acute respiratory syndrome coronavirus type 2 (SARS-CoV-2) [18]. Therefore, obesity and its comorbidities can increase the patient inflammatory status, resulting in a higher inflammation set-point and susceptibility to infection, preparing for an immune dysfunction with increased respiratory complications [19]. scWAT exhibits a greater potential than VAT to undergo beiging, a process by which white adipocytes become brown-like and participate in energy dissipation curbing obesity and related risks [20]. EAT seems intrinsically to possess a beige phenotype that may be regarded as a metabolic activation, and thus as protective against metabolic diseases. Moreover, an active thermogenic adipose tissue helps regulate circulating branched-chain amino acids, thus protecting individuals against obesity and diabetes [21]. However, it is not clear how the thermogenic induction in these particular tissues may impact cardiovascular functioning and its use for the treatment of CVD remains controversial [20,22,23]. As previously stated, it is evident instead how EAT, especially under inflammatory and metabolic insult, is capable of producing and altering adipokines, causing their shift towards a harmful phenotype [24,25].

In recent years, the interest in a precise EAT quantification is increased to potentially establish it as a reliable and non-invasive biomarker for cardiovascular risk. An expansion of EAT, namely in thickness or volume, has been related to a higher cardiometabolic risk, while its density has been proposed as a biomarker of cardiovascular risk [26]. More generally, the metabolic syndrome is linked to variations in EAT, a condition characterized by disruption of the physiological pathways regulating inflammation, given the similarities between its metabolic asset and that of visceral adipose tissue [27]. The role of EAT has been studied in patients affected by diabetes, confirming that it seems to be associated with obesity, fasting blood glucose levels, insulin resistance, and adiponectin levels in patients with type 2 diabetes mellitus (T2DM), and its increase was observed in patients with type 2 or 1 diabetes [28]. The reasons for this increase are not well identified, but possible causes include decreased physical activity and an enhanced food supply, which favor the adipose tissue deposition in diabetic patients [29]. Moreover, genes associated with lipid metabolism have been shown to be altered in diabetic patients as well as the distribution of sex steroids in body fat tissue [30]. Previous studies have shown that EAT is related to insulin levels in addition to resistin mRNA levels and may be involved in the formation of insulin resistance [31]. A significant increase has also been observed in chronic systemic inflammation, such as obesity and hyperlipidemia, and it could be also involved in abnormal lipid metabolism in the body. Regardless, the abnormal increase of EAT in diabetic patients could be used as an independent predictor of new diabetes and as a target for new therapy [32].

Different imaging techniques can study EAT and include echocardiography, cardiac magnetic resonance (CMR) and computed tomography (CT). Echocardiography is very useful to provide maximum thickness and provides a simple and easily accessible tool for EAT measurement, although it may not reflect the absolute amount of epicardial fat and gives less accurate results due to uneven distribution of epicardial fat and poor acoustic window [33]. Previous studies, however, have reported that echocardiographic measurement closely correlates with the total volume of EAT on other radiologic modalities and considering the great diffusion of this method in clinical practice, it could be very simple to obtain data on EAT [34] (Figure 2).

EAT is identified as the echo-free space between the outer wall of the myocardium and the visceral layer of the pericardium. According to the method first described and validated by Iacobellis et al., EAT thickness was generally measured perpendicularly on the free wall of the right ventricle at the end systole in 2–3 cardiac cycles. Maximum EAT thickness was measured at the point on the free wall of the right ventricle along the midline of the ultrasound beam, perpendicular to the aortic annulus [35]. Thickness varies among patients and usually ranges from 1 mm to 23 mm, but no definitive cut-off value has been identified [27]. For example, one study found a cut-off of 5.8 mm for predicting high-risk coronary plaques [36], while another reported that values of 9.5 mm in men and 7.5 mm in women could be considered thresholds to predict the metabolic syndrome [37]. On the other hand, CT and CMR can give also three-dimensional estimates of the overall EAT volume. In particular, the first shows higher spatial resolution and reproducibility but is limited by radiation exposure and long segmentation times, and the second is radiation-free but limited by lower spatial resolution and reproducibility, higher cost, and difficulties for obese patients [33,38]. However, thanks to different post-processing steps and without specific acquisitions, CT scans allow the calculation of EAT density, using Hounsfield Units (HU), both globally or on the perivascular coronary fat alone [26]. Moreover, the application of coronary CT angiography (CCTA) is rapidly increasing, following the ESC guidelines on the management of patients with suspicious coronary artery disease [39]. The availment of CCTA can provide data regarding coronary stenosis, plaque and characterization of EAT [5,40,41] (Figure 3).

EAT may be visible as a hypodense layer, with a density usually ranging from −190 to −30 HU, placed between the myocardium and the visceral pericardium. It can be quantified from different scans, most often either unenhanced scans for calcium score or CT angiographic scans, given that the EAT and pericardium are similarly visible in both, especially during the short time interval that is between contrast administration and angiographic acquisition. Therefore, adding EAT study to the output of a CT report could provide important additional information without adding to patient discomfort and radiation exposure [33]. At the same time, standard CCTA, if used only for obtaining EAT estimates, could be questioned due to the ionizing radiation exposure; however, recently Nagayama et al. proposed the use of non-gated CT, obtaining excellent concordance with gated CT but with less radiation [42]. Another limitation of EAT quantification by CT angiography is related to long segmentation times because a precise measurement is best performed on numerous small slice-thickness images; the computer software then determines EAT volume as the sum of areas of all images, accounting for slice thickness and intersection gaps. Meanwhile, several software solutions have been developed to expedite this process and rapid and fully automated algorithms for EAT volume and attenuation quantification were described [5]. La Grutta et al. conducted a study on the quantification of EAT both in coronary calcium score and CT coronary angiography image data sets comparing attenuation values, thickness and volumes obtained with these methods. The calcium score and CT coronary angiography may be used singularly in order to reduce the radiation dose, a systematic comparison of EAT assessment was useful to define their interchangeability. The image data sets may be equally employed for EAT quantification, but an underestimation of volume is found with CT coronary angiography acquisition even after post-contrast attenuation adjustment [43].

Nowadays, no globally accepted cut-off to define normal and pathological values for CT-derived EAT measures is defended. Spearman et al. reported an EAT volume above 125 mL as an indicator of cardiac pathology [44]. Other investigators have shown even that an EAT threshold between 113 and 120 cm^3^ has the greatest predictive value for future cardiovascular events [41,44]. CT allows the quantification of EAT X-ray radiodensity, which may be a biomarker of its metabolic activity, possibly related to cardiovascular or metabolic diseases [45]. Vascular inflammation localized in the coronary arteries leads to an increased risk of coronary artery disease (CAD)-related events and produces biological alterations to local cardiac adipose tissue depots. Coronary CT provides information on inflammatory changes to both EAT and PCAT, as independent markers of coronary risk. Attenuation of PCAT on CT provides indirect quantification of coronary inflammation and could be emerging as a promising imaging implement in both stable and “vulnerable” populations. While standardized CT thresholds of inflammation are yet to be established, they present a powerful avenue to enhance primary prevention initiatives [10]. A new biomarker called “fat attenuation index (FAI)”, defined as the mean PCAT attenuation within a radial distance from the outer coronary artery wall equal to the average vessel diameter, was proposed. It was higher in patients with CAD than those without CAD and was associated with coronary stenosis greater than 50%. The presence of inflammation results, in fact, in smaller pre-adipocytes with less intracellular lipid, causing an increase in PCAT attenuation [46] (Figure 4).

The purpose of this review is to analyze the state of the art literature on cardiac adiposity and its pathological correlations, with attention to gender differences [47,48]. The use of coronary CT and other imaging methods are now an integral part of daily clinical practice; thus we will summarize the knowledge on the use of EAT images such as clinical biomarkers. In particular, we will focus on the evaluation of EAT that could become a predictive/prognostic factor able to help clinicians to identify the patients at risk or not to develop cardiovascular and/or metabolic diseases, and to provide information on their clinical and therapeutic outcomes.

## 2. Materials and Methods

This review is based on focused research of the current literature on PubMed and Scopus such as research databases. We selected 2173 papers through these keywords: “epicardial adipose tissue” AND “metabolic diseases”, “cardiovascular diseases” AND “imaging” AND “prognosis”. Then, the literature was scanned selecting English papers of the last 10 years (last research of 20 March 2022), excluding reviews. Separately, two researchers MGT and CP examined and read 214 full selected papers. We included clinical trials and randomized clinical trials written in English and excluded duplicates, animal studies and studies with a lack of a group of interest, non-relevant outcomes and inappropriate methodology. Retrospective studies, case reports, case series, original articles, brief communications and letters to the editor were included in our research. We further completed our research by the examination of every study’s bibliography. We finally assessed for eligibility 50 papers with a total of 10,458 patients. Figure 5 shows the flowchart of the literature eligibility assessment process. Our review concerns the impact of the EAT measurement as a possible predictive and/or prognostic factor in patients with cardiovascular and metabolic diseases. For each paper, we analyzed all patients’ characteristics (BMI, hypertension, dyslipidemia, diabetes, smoking), therapies and pathologies putting them in relation to the EAT measurement and its correlation with clinical outcomes. Given its peculiar properties and rapid responsiveness, EAT may serve as a therapeutic target for medications or a marker to evaluate like a basal risk factor or during follow-up in patients with metabolic and/or cardiovascular diseases.

## 3. Results

The visceral adipose tissue and its cytokines act on various biological systems [6]. EAT and abdominal VAT share embryological origin from splanchnopleuric mesoderm and they secrete several inflammatory mediators but the abdominal VAT seems to have a strong correlation with metabolic risk factors, while EAT is more associated with coronary disorders [49]. Indeed, the most relevant difference to be noted is the fact that abdominal VAT produces metabolites that are distributed to the whole body through systemic circulation, so it can impact not only cardiac pathophysiology but systemic metabolic diseases as well [50]. On the other hand, EAT is smaller than abdominal VAT but directly adheres to coronary arteries; thus, it can induce significant inflammatory changes, especially on the vessels wall, and the area of EAT was larger in patients with coronary calcification compared to those without it, while the area of abdominal VAT was not [51]. However, defining the single role of the different adipose tissues on the development and progression of certain pathologies remains difficult to do. At the same time, the importance of the EAT pro-inflammatory role, studied using the routine images of the clinical practice, is grown for the predictive and prognostic information in several clinical contexts, and becomes one of the parameters to be evaluated in the diagnosis and follow-up of certain cardiovascular and endocrine-metabolic pathologies [6]. Therefore, we analyze the clinical studies on epicardial fat connected with the pathological prognosis, and to maximize comprehension, we summarized the evidence acquired in the following Table 1, Table 2 and Table 3.

### 3.1. Epicardial Adipose Tissue and Cardiovascular Diseases

The aforementioned Table 1 shows the main features of the selected studies that focused on the impact of EAT on cardiovascular diseases (study design, number and specifications of participants, methodology used for EAT evaluation and significant results regarding EAT itself).

EAT has recently been included among the parameters used to draw up a new possible machine learning to estimate cardiovascular risk, starting from a large prospective study that involved 1912 asymptomatic subjects with long-term follow-up after coronary artery calcium (CAC) scoring [52]. EAT volume and density were quantified using a fully automated deep learning method, which is incorporated into research software QFAT and provides segmentation of EAT, used to compute the EAT volume as well as the average density in HU. In this prospective study, an objective machine learning score integrating clinical data and quantitative CT assessment, after CAC scoring, appeared to be useful for the prediction of long-term risk of myocardial infarction and cardiac death. Age, atherosclerotic cardiovascular disease (ASCVD) risk score and calcium-related information were the highest contributors to prediction, but other parameters, just like the EAT volume, were also important to provide a more accurate estimation. The evaluation of the EAT should take part in clinical practice in order to facilitate physicians in assessing the cardiovascular risk of patients [52], as demonstrated in a long-term prospective cohort of 90 patients with non-obstructive coronary atherosclerosis. It was evaluated the longitudinal change in EAT volume and density on baseline CCTA performed for suspected coronary artery disease to undergo a repeat research CCTA: EAT volume increased and EAT density decreased at long-term follow-up regardless of traditional cardiovascular risk factors, age and statin use, suggesting that EAT may be considered an independent parameter rather than a surrogate for cardiovascular risk [53]. In a similar context, coronary CT was performed in 760 patients with acute chest pain. CAC score was calculated using the Agatston method while EAT volume was semiautomatically calculated. The major adverse cardiovascular events (MACE) group had a higher median of EAT and a higher prevalence of EAT > 125 mL. CAC and fat volumes were finally independently associated with MACE in acute chest pain patients, but although fat volumes might add prognostic value in patients with CAC > 400, CAC seemed to be most strongly correlated with outcome [54].

Regarding this, one of the largest studies conducted in recent years on the role of EAT has been that of Eisenberg et al. who evaluated how EAT volume and attenuation quantified by fully automated deep-learning software on non-contrast cardiac CT could predict MACE in 2068 asymptomatic subjects. At 14 ± 3 years, 223 subjects suffered MACE. Independent of gender, markers of obesity, CAC and ASCVD risk, increased EAT volume and decreased EAT attenuation were both independently associated with MACE, providing a very important prognostic value to these measures. In particular, EAT volume ≥ 113 cm^3^ and EAT attenuation ≤ −77.0 HU were associated with a greater risk of adverse outcomes, with the highest MACE risk in subjects with EAT volume ≥ 113 cm^3^ and CAC ≥100 AU. Notably, in this study, EAT predicted MACE in patients with no coronary calcium and also correlated with inflammatory circulating biomarkers, identifying subjects at increased risk for future events and in whom lifestyle modification, statin therapy, and anti-inflammatory therapy should be initiated [41].

To underline the impact of a possible anti-inflammatory therapy on EAT, it is certainly fair to mention the study by Almeida et al., the first to investigate changes in EAT with an anti-inflammatory drug, a potent 5-lipoxygenase inhibitor (VIA2291). Coronary CT angiography was performed at baseline and 24 weeks after treatment, and then have been analyzed 54 pre- and post-treatment scans (patients with myocardial infarction or unstable angina 21 days before randomization). There was a significant decrease in EAT volume in patients in the treatment arms versus placebo, suggesting also a possible dose-dependent effect because of an incremental reduction in EAT with higher doses of VIA-2291. Moreover, a statistically significant correlation between the change in EAT and total plaque volume was found, so treatment also altered plaque morphology primarily through a reduction of the fibrocalcified component of atherosclerotic plaque. This study confirmed how adipose tissue can act as a reservoir for pro-inflammatory cytokines influencing the development of atherosclerotic plaque [55].

Even statin therapy seemed to impact EAT characterization. In a population of postmenopausal women with subclinical atherosclerosis and hypercholesterolemia, a moderate to aggressive treatment with statins reduced the EAT attenuation on CT scan by 6% after 1 year of treatment [45]. Moreover, intensive atorvastatin therapy (80 mg/day) in atrial fibrillation patients who underwent pulmonary vein isolation was associated with significant decreases in EAT assessed by cardiac CT at baseline and after 3 months [56]. At the same time, since also a reduction of the protective BAT activity is correlated to a decrease of the CT attenuation of EAT [100,101], it could reflect even inhibition of BAT physiological features giving a negative influence on statins, which obviously have anyway a proven record of cardiovascular safety. However, if the inhibition of BAT and browning of WAT correlate to worse outcomes in patients receiving statins remains to be studied in detail, indeed it is possible that BAT itself may become pro-atherogenic in some disease states, such as hypercholesterolemia and obesity [45]. 

Regardless, a positive impact that the lifestyle alone (correct nutrition and regular activity) can have on EAT has also been highlighted by other recent studies. One demonstrated that improving nutritional quality and being physically active could decrease cardiometabolic risk markers through changes in visceral/ectopic fat depots that are not reflected by changes in body weight alone [57]. Another tried to investigate how specific dietary supplementation could impact fat levels, including those of EAT. Based on proven evidence that garlic is able to give potential cardiovascular benefits by retarding the progression of coronary atherosclerosis, lowering cholesterol levels and blood pressure, reducing platelet aggregation and adhesion, preventing low-density lipoprotein oxidation and suppressing atherosclerosis [102,103,104,105], authors studied the effect of aged garlic extract with B-vitamins, folic acid, and L-arginine on change in intrathoracic and subcutaneous adipose tissue depots over a -1year period, measured with CAC scans. A beneficial effect of this supplementation on fat volumes emerged and remained significant even after adjusting for cardiovascular risk factors, statin therapy and BMI [58]. Similar benefits were also observed by Ahmadi et al. [59].

An important study that tried to analyze the predictive value of the EAT was conducted by Maimaituxun et al. with the aim to measure the local EAT thicknesses surrounding the coronary arteries and determined their clinical utility for predicting CAD. A total of 197 patients underwent 320-slice multi-detector coronary angiography CT and were segregated into CAD (≥1 coronary artery branch stenosis ≥50%) and non-CAD groups. The local fat thickness surrounding the left anterior descending (LAD) artery was a simple and useful marker for estimating the presence, severity, and extent of CAD, independent of classical cardiovascular risk factors [60].

The evaluation of the EAT could provide interesting information even about the characterization of atherosclerotic plaques, helping to identify those at greatest risk of rupture [106]. In a cohort of 467 subjects with acute chest pain who had both non-contrast CT and CTA, was found a significant association between greater EAT volume (indexed EAT volume > 62.3 cc/m^2^) and high-risk plaque features, also adjusting for potential confounders such as traditional cardiovascular risk factors, CAC score, and coronary artery stenosis. Moreover, lower attenuation of EAT appeared associated with high-risk plaque features. So, these findings supported the possible local influence of EAT depots on coronary atherogenesis [61], especially considering that even in a cohort of patients with a CAC score of zero, a high EAT volume was associated with the presence of non-calcified coronary plaques on CCTA with low VAT area [62]. Interesting to note that EAT had an increased echocardiographic thickness and was a source of inflammatory mediators also in patients with calcific aortic stenosis, supporting the hypothesis of an involvement of cardiac visceral fat in inflammatory and atherogenic phenomena occurring in the aortic valve and promoting its degeneration and calcification [63].

EAT appears to play a role also in the development of cardiac arrhythmias. 116 Asian patients who had undergone dual-source CT and a 24-h Holter ECG were retrospectively enrolled to evaluate the potential impact of pericardial fat on ventricular arrhythmia. Authors found that it was significantly associated with the occurrence of ventricular premature beats, in particular the specific component of right ventricular pericardial fat [64]. Moreover, EAT volumes measured using CT were significantly associated with atrial fibrillation recurrence after catheter ablation [107,108]. At the same time, echocardiographically measured preprocedural EAT seemed to predict this recurrence rate after cryoablation positively correlating with C-reactive protein as an indicator for systemic inflammation. Therefore, this is another medical area where EAT could be useful as a guide in clinical practice, specifically to identify patients at high risk of arrhythmia recurrence, providing them closer follow-up [65].

The coronary perivascular adipose tissue was increased at the spastic coronary segment of vasospastic angina patients, suggesting its involvement even in the pathogenesis of coronary spasm. In particular, PVAT volume in the LAD coronary arteries on CT coronary angiography appeared significantly increased in 48 patients with LAD spasms compared with 18 controls [66].

### 3.2. Epicardial Adipose Tissue and Metabolic Diseases

The aforementioned Table 2 shows the main features of the selected studies that focused on the impact of EAT on metabolic diseases (study design, number and specifications of participants, methodology used for EAT evaluation and significant results regarding EAT itself).

Over the years, many studies have focused on the role of an adequate lifestyle as a first-line therapy for metabolic diseases and specifically investigated its possible impact on the EAT. Christensen et al. randomized 39 physically inactive participants with abdominal obesity to a supervised high-intensity interval endurance training (3 times a week for 45 min), resistance training (3 times a week for 45 min), or no exercise (control group), evaluating changes in epicardial and pericardial adipose tissue mass assessed by CMR. It emerged that both endurance and resistance training reduced EAT mass, while only resistance training reduced pericardial adipose tissue mass. Nowadays, different pharmacological strategies, such as new antidiabetic therapies [68,69], seem to impact cardiac adipose tissue reduction, but getting the same only with correct physical activity would certainly be a great advantage, retaining it as a potential preventive strategy [70]. Similar results were obtained during a pilot study using CMR to quantify cardiac fat in which 3-week high-intensity, moderate-volume muscular endurance resistance training was able to reduce EAT and PAT volume and improve body composition, muscular strength, and cardiorespiratory fitness in 11 young females with obesity, with no negative effects on arterial stiffness [71]. Even a 12-week resistance circuit training program (3 days per week) reduced EAT in 48 obese aged women and was effective and safe [72].

In particular, lifestyle intervention is an essential first-line strategy for clinical management of metabolic syndrome, improving its prevalence and individual components as well as cardiovascular health. Physical exercise is a well-established non-pharmacological intervention for the prevention and treatment of several chronic and cardiovascular diseases. EAT appeared significantly higher in patients with metabolic syndrome and its reduction was independently associated with improvements in cardiac parameters in obese individuals [73,109]. It has been demonstrated baseline EAT significantly increased in patients with metabolic syndrome compared to controls, and its reduction over time could partly explain the myocardial function improvements observed following lifestyle intervention (restrictive diet combined with an aerobic plus resistance training program), thanks to a limited myocardial and coronary artery inflammation with a decreased tissue oxidative stress [74].

Although aerobic physical exercise is the first-line treatment for metabolic syndrome with elevated adipose accumulation, there are other few data on exercise-induced changes in EAT in these patients. Jo et al. evaluated the impact of exercise training on the morphology of the EAT, in a single-center study that included 34 patients with hypertension and metabolic syndrome, using standard two-dimensional echocardiography to estimate the EAT thickness. Compared to moderate-intensity continuous training (MICT), high-intensity interval training (HIIT) appeared to be a time-efficient exercise strategy for cardiometabolic health. EAT decreased significantly in both the groups, but patients in the HIIT group showed a greater reduction, maybe partly attributed to the greater elevation of post-exercise metabolic rate and associated fat expenditure in HIIT. Therefore, these results suggested the possibility to recommend HIIT for patients with hypertensive metabolic syndrome to have a good improvement in EAT and endothelial function [75]. Even a supervised home-based 16-week treadmill training program appeared useful in postmenopausal women with metabolic syndrome, resulting in an EAT reduction [76].

Type 2 diabetes mellitus is a major risk factor for CVD, but a simple reduction in blood glucose levels is not sufficient for cardiovascular protection. It has been suggested that cardiac fat accumulation causes excessive release of pro-inflammatory cytokines and free fatty acids, resulting in myocardial intracellular lipotoxicity, myocardial fibrosis and cardiac dysfunction. Mohar et al. found an independent association between EAT volumes with the severity of CAD in an asymptomatic cohort of patients with T2DM after adjustment of traditional risk factors, BMI, and CAC scores. EAT volumes could predict the presence and severity of CAD independent of CAC, representing an objective imaging marker in the assessment of CAD severity in this patient category [77,78]. 

Lifestyle changes, with proper diet and regular physical activity, are certainly the first weapons to be used in the management of diabetes [79]. Leroux-Stewart et al. conducted a pilot randomized controlled trial on 73 type 2 diabetic patients in which, in a context of a similar prescribed caloric restriction diet, adding physical activity allowed larger fat mass and epicardial fat thickness reductions, involving the reduction of cardiovascular risk. Two-dimensional transthoracic echocardiography has been used to measure the epicardial fat thickness. The caloric restriction strategy induces a 7.5% reduction in epicardial fat thickness whereas the combination has a 21% decrease. The intrinsic fast metabolism of EAT could make it a good responder to interventions targeting fat reduction. Moreover, the greater EAT loss in a combined group could be attributed to increased heart rate experienced during exercise, resulting in higher blood flow and lipolysis in the EAT with its marked reduction [80]. During a 16-week low-calorie diet, it reduced body weight, pericardial fat, hepatic triglyceride content, visceral and subcutaneous abdominal fat volumes compared with baseline values. After an additional 14 months of follow-up on a regular diet, the reduction in pericardial fat volume was sustained, despite a substantial regain in body weight, visceral abdominal fat, and hepatic triglyceride content in 14 obese patients with insulin-treated T2DM, in which pericardial fat was measured using MR imaging and proton spectroscopy [81].

EAT was reported to be correlated with obesity and diabetes [28]. Therefore, obese patients with T2DM have an extremely high risk of developing ectopic adiposity. SGLT2 inhibitors reduce body fat, including visceral fat, and cardiovascular events in these patients [110]. Therefore, it would be very interesting to know if they are able to reduce specifically EAT. Luseogliflozin could reduce the EAT volume in parallel with the improvement of systemic micro-inflammation and the reduction of body weight in Japanese patients with T2DM and BMI ≥ 25 kg/m^2^, assuming the possibility that SGLT2 inhibitors may impact cardiovascular risk partly by reducing the EAT volume. The reduction of EAT was measured by CMR at 12 weeks and significantly correlated even with the reduction of C-reactive protein [82]. Likewise, the dapagliflozin addition to metformin monotherapy significantly reduced ultrasound-measured EAT thickness in patients with T2DM and overweight/obesity, giving this effect early because it occurred just after 12 weeks of treatment, partially independent of weight loss [83]. Similar results have been obtained with the use of liraglutide, an analog of glucagon-like peptide-1, probably able to reduce cardiovascular risk [111], although is not known if these beneficial effects could be attributed to drug effect on visceral fat. Ultrasound-measured EAT thickness was measured at baseline and 3- and 6-month follow-ups in 85 subjects with T2DM and overweight/obesity, and the liraglutide added to metformin group has been documented as a 29% and 36% of reduction at 3 and 6 months, providing a potential explanation on the mechanisms behind the cardioprotective effects of liraglutide [69]. The GLP-1 receptor is expressed in the adipose tissue and mRNA and protein expressions are increased in visceral fat. The drug-induced browning effect on EAT certainly warrants further investigations, but it also has been suggested that GLP-1 promotes preadipocyte differentiation, improving local insulin resistance [112,113,114]. Even exenatide seemed to be an effective treatment to reduce liver fat content and EAT in obese patients with T2DM. Using CMR, a 9% loss of EAT in the exenatide group was observed after 26 weeks of treatment [84]. In a non-randomized cohort of 25 people with T2D, Morano et al. reported a significant reduction in the thickness of EAT (−15%) after 3 months of GLP-1-RA treatment using echocardiography as an imaging method [85].

Fortunately, in recent years, many new drugs have been identified in the management of diabetic patients, and although both SGLT2 inhibitors and DPP-4 inhibitors seem to reduce EAT [115,116], it is unclear whether SGLT2 inhibitors are superior to DPP-4 inhibitors in reducing cardiovascular or cardiometabolic risk factors in patients with early-stage T2DM, without CVD and with a good cardiac function. Therefore, a recent prospective randomized study on 42 diabetic patients demonstrated that 12-week administration of empaglifozin had similar effects as sitagliptin on cardiac fat accumulation, evaluated with CMR, and cardiac function in patients with early-stage T2DM, without CVD complications. However, regarding cardiometabolic biomarkers, early supplementation with SGLT2 inhibitors may be preferable to DPP-4 inhibitors to provide early cardiac protection and primary prevention of CVD [79]. It was investigated also the effect of 2 distinct insulin basal analogs, detemir and glargine, on total body weight, body composition and EAT thickness among insulin-naïve patients with T2D and poor glycemic control. After 6 months, detemir had minimal effect on changes in body weight while glargine resulted in significant weight gain. EAT significantly decreased from baseline in both groups but detemir had a trend for a more pronounced echocardiographic EAT thickness reduction [86].

As previously explained, EAT is known to be involved not only in the pathogenesis of coronary artery disease but also in the development of arrhythmogenesis. A study investigated the effect of SGLT2 inhibitors dapagliflozin on EAT volume and P-wave indices. 35 patients with T2DM and coronary artery disease were classified into dapagliflozin group and conventional treatment group, and EAT volume was measured using electrocardiography-gated cardiac CT scans. Dapagliflozin has been shown to reduce plasma levels of TNF-α, EAT volume, and P-wave indices and dispersion. The changes in P-wave indices were especially associated with changes in EAT volume, testifying to the fact that EAT may lead to arrhythmogenesis via structural and electrical remodeling of the heart chambers (direct fatty infiltration, fibrosis of the atrial myocardium through the secretion of adipo-fibrokines member of the TGF-β superfamily, local inflammatory processes supported above all by TNF-α). Moreover, the decrease in EAT thickness also improved obesity-related cardiac morphological and functional changes during body weight loss, observed especially in the dapagliflozin group [68].

Interestingly, in the diabetic patient population, some studies have attempted to identify some surrogate markers for evaluating atherosclerosis that are non-invasive and low-cost, unlike others which are useful but somewhat inconvenient and expensive to obtain, such EAT volume itself. A close relationship between EAT accumulation, quantified on ECG-gated diagnostic cardiac CT scans, and the serum level of cystatin C, independent of glomerular filtration rate, has been demonstrated in a population of diabetic patients [87]. EAT has distinctive pathogenic and pathophysiological characteristics with exacerbate inflammation and oxidative stress and cystatin C might be expressed in EAT and secreted into the circulation. Moreover, elevated cystatin C seems to have a link with the development of CVD in subjects without chronic kidney disease (CKD), and its level showed a strong correlation with the degree of CAD and all-cause mortality [117,118,119]. Therefore, these results could indicate that cystatin C is not simply a marker of impaired kidney function but also a marker of cardiovascular pathologies. Considering that individuals with T2DM have elevated levels of plasminogen activator inhibitor-1 (PAI-1), implicated in the pathophysiology of CVD [120,121], and visceral adipose tissue is metabolically more active than subcutaneous adipose tissue producing more PAI-1 [122], a similar study was conducted on 51 individuals with well-controlled T2DM and no prior history of CVD investigating the relationship between two cardiovascular risk factors. It demonstrated that PAI-1 levels positively correlate with a pericardial fat volume estimated with CMR [88], while another one showed an association between the specific epicardial fat and PAI-1 in 27 obese women without diabetes [123].

### 3.3. Epicardial Adipose Tissue in Other Clinical Conditions

The aforementioned Table 3 shows the main features of the selected studies that focused on the impact of EAT in some specific clinical conditions (study design, number and specifications of participants, methodology used for EAT evaluation and significant results regarding EAT itself).

In the subpopulation of CKD patients, it has been shown a significantly increased coronary CT EAT volume compared with non-CKD patients, and that was associated with high-risk coronary plaque [89]. A study conducted by Yazbek et al. evaluated the role of EAT volume in the kidney transplantation population and its association with CVD, considering that obesity is common in patients with CKD and weight gain is observed in approximately one-third of patients after kidney transplantation. The EAT progressor group was defined by any increment in EAT after 12 months, using thoracic CT. No relationship was observed between the presence or progression of coronary calcification and EAT, but this result could be attributed to the young age of the study population, short duration of previous dialysis therapy, low prevalence of diabetic patients and incidence of coronary calcification, and a well-functioning kidney graft [90]. To date, baseline measurement of EAT has been associated with all-cause mortality in a single study of hemodialysis patients [124] and with cardiovascular events in CKD 3–5 patients [125], becoming an indicator of early atherosclerosis in hemodialysis patients [91]. At the same time, little knowledge of the prognostic impact of EAT progression is present. Sevelamer is a non-absorbable polymer indicated for hemodialysis patients as a phosphate-binding agent and reduces the serum level of cholesterol and inflammatory markers, slowing the progression of coronary artery calcium with a lower mortality rate compared to calcium-based phosphate binders. Although the difference between treatments was not statistically significant, it has been demonstrated that EAT progression from baseline was smaller with Sevelamer than with another calcium-based phosphate binder in incident hemodialysis patients, after 18 months of follow-up with coronary CT [92].

Another factor that could affect the progression of EAT is the continuous positive airway pressure therapy (CPAP) used in patients with obstructive sleep apnea (OSA), a pathology very common in patients with metabolic diseases. Although the mechanisms involved are not well understood, in a cohort of symptomatic patients with AHI > 15 who received CPAP therapy for 24 weeks, there was a significant decrease in EAT values evaluated with two-dimensional transthoracic echocardiography. Stratifying OSA degree, EAT appeared significantly higher in patients with AHI > 15, highlighting a relationship between EAT and OSA severity [93], as previously noted [126,127].

EAT was recently assessed in menopausal women, precisely because endogenous estrogen may regulate these fat depots and increase after menopause [128]. It was interesting to note how hormone replacement therapy impacts differently on the same according to type, combination, dose, and route of administration: therapy with oral conjugated equine estrogens may slow EAT accumulation, whereas transdermal 17β-estradiol may increase the progression of coronary artery calcification associated with fat accumulation, among 474 early menopausal women in which heart fat depot volumes were measured using existing CT scans before randomization (baseline) and 48 months after. The stronger antioxidant property of ring B unsaturated estrogens present in oral therapy likely contributes to its favorable impact on EAT accumulation in this study. A weak antioxidant mechanism, together with overexpression of oxidative stress could contribute to adiposity-related complications [94].

Indeed, the exercise training reduced the amount of visceral fat, EAT obtained by CMR imaging, in patients with major depressive disorder, improving factors constituting the metabolic syndrome. Thus, given the high prevalence of cardio-metabolic disorders in major depression, exercise training has to be recommended as an additional treatment in this patient category [95].

Few studies have focused on the assessment of EAT in children and their field of investigation was very diversified. Using docosahexaenoic acid supplementation in overweight children with NAFLD seemed to decrease liver and visceral fat (including EAT), ameliorating metabolic abnormalities [96]. EAT thickness was studied even in children with subclinical hypothyroidism with the aim to identify any relation to early subclinical atherosclerotic changes. The authors analyzed clinical, laboratory and ultrasound parameters, such as carotid intima-media thickness (CA-IMT), brachial artery flow-mediated dilation (FMD) responses and EAT thickness. They demonstrated that EAT was higher in children with subclinical hypothyroidism compared with controls and associated with FMD responses. It could be considered an additional marker of endothelial dysfunction in this subgroup of children, and as TSH stimulates adipogenesis from an early stage, its higher levels could be associated with a preferential increase in EAT, thus contributing to increased cardiovascular risk [97]. Interestingly, Çelik et al. first reported that lean children with premature adrenarche had increased EAT (obtained with echocardiographic investigations) compared to age- and sex-matched controls, and increased EAT was also positively correlated with DHEA-SO4 levels [98].

The cardiac adipose tissue volume and density appeared to be independently related to insulin resistance at baseline but did not predict changes in insulin resistance over time in a placebo-controlled trial of rosuvastatin in patients with treated HIV infection. In particular, after adjustment for BMI, the relationship of HOMA-IR with fat volume was attenuated, but density remained associated [99].

## 4. Discussion

This review included 50 studies that contained 10,458 patients, which can be grouped into three broad study categories based on the prevalent pathologies: cardiovascular disease, metabolic diseases and other clinical conditions. Given the increasing use of coronary CT and imaging methods capable of identifying and quantifying epicardial fat, we have tried to analyze the impact of EAT on patient outcomes. Although some different imaging techniques can study EAT, each shows its advantages and disadvantages. Echocardiography is readily available, inexpensive, quick, but operator-dependent, unable to deliver volumetric measurements and difficult in patients with motion or breathing issues. CCTA provides high definition, volumetric measurements, best visibility of the pericardium, but even long segmentation times and ionizing radiations. CMR allows volumetric measurements and quantification of attenuation without ionizing radiations but is expensive in terms of time and cost, not always available, able to hardly visualize pericardium and is contraindicated in some patients. However, the application of coronary CT is rapidly increasing, following the ESC guidelines in the management of patients with suspicious coronary artery disease. Therefore, this imaging method could provide EAT as a standard additional measure for all patients undergoing examination.

Several studies showed that the evaluation of the EAT should take part in clinical practice in order to facilitate physicians to assess the patient’s cardiovascular risk. Increased EAT volume was a predictor of the presence of CAD, acute myocardial infarction, and “high-risk” CAD phenotypes. EAT density or attenuation has likewise been associated with CAD in numerous observational studies, but the broad nature of this marker remains to be specifically defined. EAT may be considered an independent risk factor rather than a surrogate for cardiovascular risk. An increased EAT volume and a decreased EAT attenuation seemed to be both independently associated with MACE, providing a very important prognostic value [41,52,53,54]. In particular, the local fat thickness surrounding the LAD coronary artery was a simple and useful parameter for estimating the presence, severity, and extent of CAD [60], and the coronary perivascular adipose tissue appeared to be increased at the spastic coronary segment of vasospastic angina patients [66].

As well as on the development of cardiovascular diseases, EAT evaluation could provide specific information even about the characterization of atherosclerotic plaques, helping to detect those at greatest risk of rupture: greater EAT volume and lower EAT attenuation were associated with high-risk plaque features, thus identifying patients who must undergo a close follow-up in view of the greater risk of rupture and ischemia [61]. Likewise, EAT could promote aortic valve degeneration and calcification [63].

EAT appeared to play a role also in the development of cardiac arrhythmias and was significantly associated with the occurrence of ventricular premature beats [64] and atrial fibrillation recurrence after catheter ablation [107,108].

Considering the impact that epicardial fat can have on the cardiovascular structure, it is important to highlight that there are different strategies that seem to act on its reduction. Although the specific mechanisms of action have not yet been well identified, anti-inflammatory therapy, statins and lifestyle alone could change positively the EAT characterization [45,55,57]. Effectively, the correlation between EAT and response to certain pharmacological therapies has been investigated. For example, statins have been shown to decrease EAT attenuation without changes in serum lipid in a cohort of patients with limited CAC [45], while a prospective study evaluating patients with subclinical CAD found no impact of statins on EAT attenuation [53]. This variance may be owed to the nature of EAT which encompasses a wide range of adipocytes of varying proximity to the vessel wall. Therefore, it is possible that inflammatory changes are more marked and clear especially in the adipocytes which are located closer to the vessels. This emerging evidence should suggest statins could have not only a lipid-lowering effect with maximum action in the liver but also a potential anti-inflammatory and antioxidant effect exerting beneficial metabolic effects in other tissues [56]. One work by Ko et al. conducted on patients undergoing hemodialysis treated with different drugs, found a lack of EAT increase during the treatment with sevelamer, possibly due to its anti-inflammatory effect [92]. Even the CPAP therapy used in patients with OSA, a very frequent symptom in patients with metabolic pathologies, seemed capable of determining changes at the level of the EAT. Although the mechanisms involved are not well understood, in symptomatic patients who received CPAP, there was a significant decrease in EAT values which showed also a relationship with OSA severity [93].

Regarding endocrine-metabolic pathologies (metabolic syndrome, obesity, diabetes), many studies have focused on evaluating the impact of therapies on EAT, including it among those factors that could be monitored during patient follow-up. For example, baseline EAT has been shown to be significantly increased in patients with metabolic syndrome, and lifestyle intervention could determine its reduction over time, partly explaining the myocardial function improvements observed in these patients [74].

An independent association between EAT volumes with the severity of CAD in an asymptomatic cohort of patients with T2DM, after adjustment of traditional risk factors, was conducted [77]. Thus, EAT could become a new therapeutic target in the treatment of metabolic-related cardiac diseases. SGLT2 inhibitors and GLP-1-RA, new drugs more and more used in the management of diabetic patients, could reduce the EAT volume in parallel with the improvement of systemic micro-inflammation and the reduction of body weight [82,84,111]. Early supplementation with SGLT2 inhibitors seemed to be preferable to DPP-4 inhibitors in providing early cardiac protection and primary prevention of CVD [79].

Interestingly, in diabetic patients, some studies have also tried to identify some non-invasive and low-cost surrogate markers for evaluating atherosclerosis which could be related to the EAT volume itself. Cystatin C and PAI-1 levels positively correlated with EAT volume [87,88] and could be used as markers of cardiovascular pathologies.

Although, as analyzed in recent years, many studies have focused on the study of EAT, many of these maintain a purely “cardiological” and less “metabolic” approach. They also carry out a local analysis related to epicardial fat without considering the systemic endocrine action that the visceral fat with its cytokines could carry out specifically in the heart. Defining the single impact of the different adipose tissues on specific pathologies remains difficult to do, but EAT is smaller than abdominal VAT, and its close and direct proximity to the heart vessels, could induce significant inflammatory changes, especially on these vessels compared to abdominal VAT. However, it is important to underline that most of the studies do not use DEXA (Dual X-ray Absorptiometry) to characterize the body composition, so the value of the total fat mass and its distribution are not identified. Other considerations that are lacking are those relating to the thyroid and adrenal axis. Indeed, in the analyzed populations, there could be patients with alterations in thyroid function that may alter the distribution of body fat affecting the evaluation of EAT in an unrecognized way. Therefore, it is always important to specify that the analyzes were performed on euthyroid patients, especially considering that a study about EAT in children with subclinical hypothyroidism showed EAT was higher in these children compared with euthyroid controls [97]. Regarding cortisol levels, especially in patients subjected to glucocorticoid therapies for long periods, is possible a similar situation precisely because also this hormone can impact and modify body composition. Moreover, in the large group of cardiovascular diseases, hypertension is certainly evaluated as a parameter and risk factor in most studies, but it was considered among the descriptive characteristics of the population, rather than being directly related to EAT characterization.

The thickness of EAT could be a useful new clinical and therapeutic target, based on the evidence gained. To underline the great beneficial potential of cardiac adipose tissue, it is necessary to cite also the AGTP II trial in which was evaluated the efficacy of the adipose graft transposition procedure in 108 patients with non-revascularizable myocardial infarction and myocardial scar. A population of human adult mesenchymal-like cells derived from cardiac adipose tissue (cardiac adipose tissue mesenchymal stem cells—cATMSCs) was identified and characterized. These cells, despite residing in an adipocytic environment, have an inherent cardiac-like phenotype and may play a role in heart homeostasis and act as a cellular reservoir for myocardial tissue renewal promoting neovascularization, improving local microcirculation, metabolism, microenvironment at the scar site, and enhancing myocardial contractility [67]. Finally, metabolic and cardiovascular diseases are multifactorial disorders to which contributes the inflammation of the adipose tissue and EAT in our specific contest. Thus, EAT be at the same time both a risk factor to be stemmed in order to prevent them and a specific target on which to act and to monitor during the clinical history of the most “vulnerable” patients. Different interventions have been shown to exert positive effects on these diseases, at least in part by modulating EAT inflammation, acting both on cytokine storm and on fat and immune cells. Certainly, further investigation is required to delineate the exact role of EAT and the mechanisms involved in carrying out its harmful role, and this could allow the emergence of novel therapeutic strategies aimed at immunomodulating the EAT.

## 5. Conclusions

The EAT has an important and peculiar secretion prorogating a chronic inflammation involved in the development of different pathologies, but before drawing definitive conclusions, it is useful to make some considerations. Many studies analyzing the role of EAT have a cross-sectional design and include small study groups and too short follow-ups to allow definitive generalizations. There are few prospective studies, different methods are used to estimate EAT without specific cut-offs correlated with diagnosis or prognosis, and sometimes, EAT measurement is not the primary study intent. However, by evaluating all the studies analyzed as a whole, EAT seems to emerge as a potential imaging biomarker, especially useful for predicting CAD and its complications, and monitoring patients with metabolic diseases during clinical follow-up. This, along with the ever-increasing use in daily practice of imaging techniques capable of evaluating it, may open the road to the possibility of using EAT as a novel, independent risk biomarker in several clinical settings. EAT could be correlated with clinical outcomes and act as a therapeutic target for new medications or a marker to investigate during the patient medical history. Thus, EAT could become a new marker for diagnostic, therapeutic and follow-up targets in the metabolic-related cardiac diseases.

## Figures and Tables

**Figure 1 nutrients-14-02926-f001:**
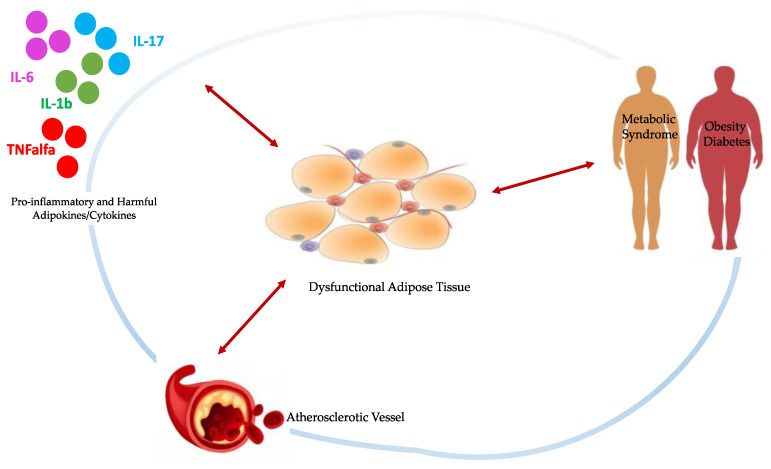
Schematic representation of bidirectional signaling between dysfunctional EAT and damaged vasculature, pro-inflammatory cytokines and stressful/pathological conditions, all interconnected by subtle cause-and-effect relationships. Physiologically, EAT contributes to maintaining vessels homeostasis through the production of anti-inflammatory adipokines and cytokines in response to the same markers of oxidative stress and inflammation from the vasculature. When stressful conditions take over, the EAT becomes hypertrophic and dysfunctional, releasing detrimental substances which contribute to the onset and progression of atherosclerosis.

**Figure 2 nutrients-14-02926-f002:**
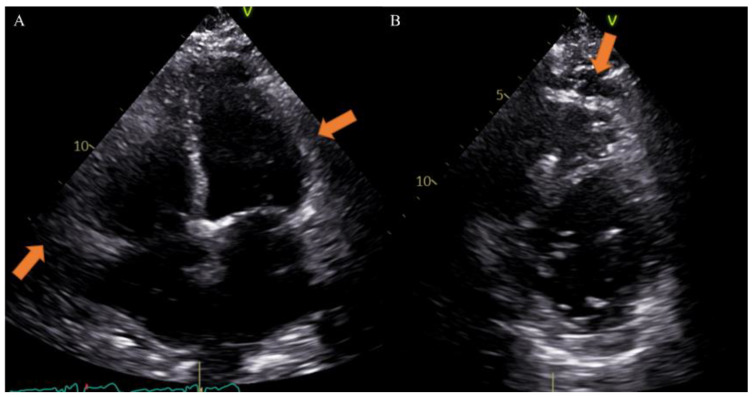
Case of a 65-year-old male patient underwent to echocardiography showing epicardial fat tissue on four chambers ((**A**), arrow) and short axis ((**B**), arrow) views. *From Giuseppe Muscogiuri’s private archive of unpublished cardiac imaging*.

**Figure 3 nutrients-14-02926-f003:**
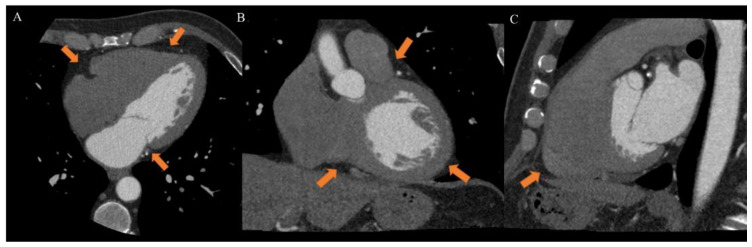
Case of a 46-year-old male patient underwent to cardiac computed tomography angiography (CCTA) for atypical chest pain. CCTA showed epicardial fat on axial ((**A**), arrow), coronal ((**B**), arrow) and sagittal ((**C**), arrow) planes. *From Giuseppe Muscogiuri’s private archive of unpublished cardiac imaging*.

**Figure 4 nutrients-14-02926-f004:**
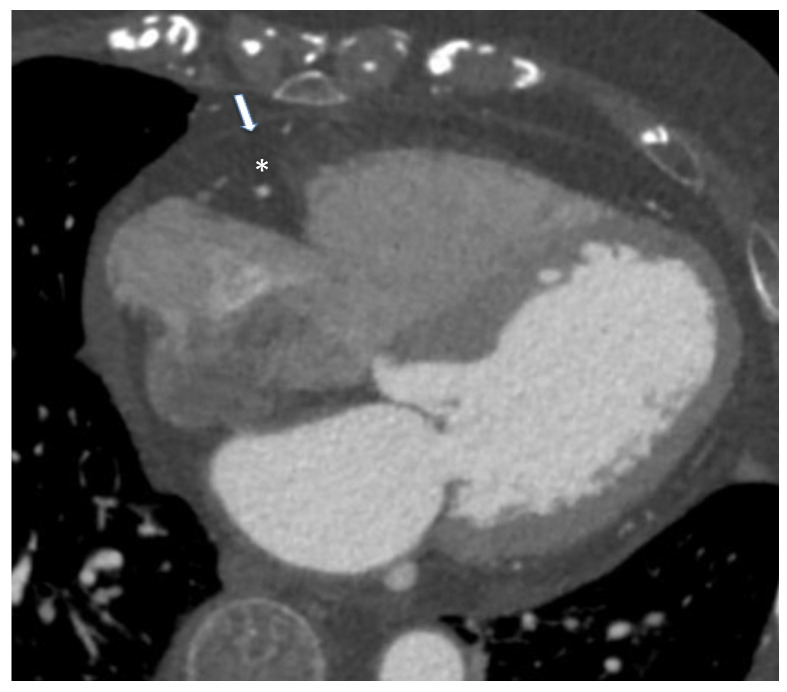
Case of an 85-year-old male patient underwent cardiac computed tomography angiography for the evaluation of coronary artery disease. Arrow identifies pericardium while asterisk represent epicardial adipose tissue. *From Giuseppe Muscogiuri’s private archive of unpublished cardiac imaging*.

**Figure 5 nutrients-14-02926-f005:**
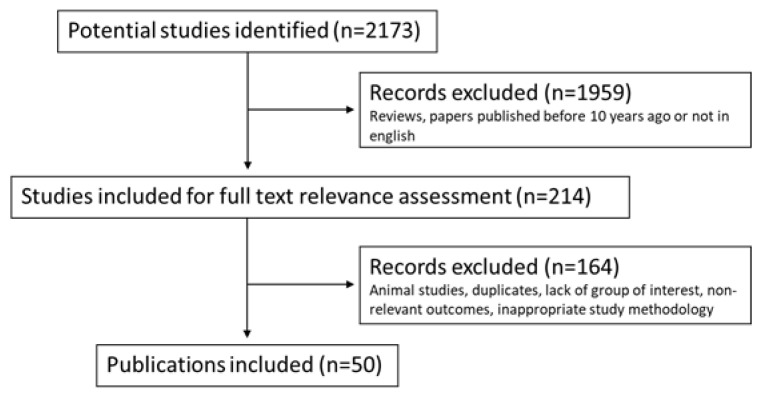
Flowchart of the literature eligibility assessment process.

**Table 1 nutrients-14-02926-t001:** Epicardial adipose tissue evaluation in cardiovascular diseases.

Study	Design	N. Pts	Population	Imaging Method	EAT Evaluation
Eisenberg et al. [41]	Prospective (FU over 14 years)	2068	Asymptomatic subjects	CT	Increased EAT volume and decreased EAT attenuation were both independently associated with MACE
Raggi et al. [45]	RCT (FU 1 year)	420	Postmenopausal women with atherosclerosis and hypercholesterolemia	CT	Statins reduced the attenuation of EAT
Commandeur et al. [52]	Prospective (FU over 14 years)	1912	Asymptomatic subjects	CT	An objective machine learning score evaluating EAT was useful for the prediction of long-term risk of myocardial infarction and cardiac death
Nerlekar et al. [53]	Prospective (FU over 4 years)	90	Patients with non-obstructive coronary atherosclerosis	CT	EAT demonstrated significant longitudinal changes with an increase in volume and decrease in density
Forouzandeh et al. [54]	Prospective (FU over 4 years)	760	Patients with acute chest pain	CT	EAT volume was independently associated with MACE
Almeida et al. [55]	Prospective (FU 24 weeks)	54	Patients with myocardial infarction or unstable angina	CT	A 5-lipoxygenase inhibitor reduced EAT with a correlation between change in EAT and total plaque volume
Soucek et al. [56]	Prospective (FU 3 months)	79	Atrial fibrillation patients who underwent pulmonary vein isolation	CT	An intensive atorvastatin therapy was associated with a decrease in EAT
Gepner et al. [57]	RCT (FU 18 months)	278	Sedentary adults with abdominal obesity or dyslipidemia	CMR	Improving nutritional quality and being physically active could decrease cardiometabolic risk through changes in visceral fat depots, like EAT
Zeb et al. [58]	RCT (FU 1 year)	60	Asymptomatic subjects	CT	Aged garlic extract with supplement could determine a decrease in EAT
Ahmadi et al. [59]	RCT (FU 1 year)	60	Asymptomatic subjects	CT	Aged garlic extract with supplement could determine a decrease in EAT
Maimaituxun et al. [60]	Cross-sectional	197	CAD patients vs. non-CAD patients	CT	The local fat thickness surrounding the LAD artery was a marker for estimating the presence, severity and extent of CAD
Lu et al. [61]	Cross-sectional	467	Patients with suspected acute coronary syndrome	CT	A greater volume of EAT was associated with high-risk coronary plaques
Tsushima et al. [62]	Cross-sectional	352	Patients with suspected coronary artery disease	CT	A high EAT volume was associated with the presence of non-calcified coronary plaques
Parisi et al. [63]	Cross-sectional	139	Patients with severe, isolated, calcific aortic stenosis	Echocardiography	EAT could promote degeneration and calcification of the aortic valve
Tam et al. [64]	Retrospective	116	Patients with suspected coronary artery disease	CT	EAT could promote the occurrence of ventricular premature beats
Canpolat et al. [65]	Prospective (FU over 20 months)	234	Patients with symptomatic atrial fibrillation subjected to cryoablation	Echocardiography	Preprocedural EAT seemed to predict atrial fibrillation recurrence rate after cryoablation
Ohyama et al. [66]	Cross-sectional	66	Patients with LAD spasm vs. healthy controls	CT	PVAT had a role in the pathogenesis of coronary spasm
Gastelurrutia et al. [67]	RCT (FU 1 year)	108	Patients with non-revascularisable myocardial infarction	CMR	A population of human adult mesenchymal-like cells derived from EAT could act as a cellular reservoir for myocardial tissue renewal

EAT: epicardial adipose tissue, MACE: major adverse cardiovascular events, LAD: left anterior descending, CAD: coronary artery disease, PTS: patients, PVAT: perivascular visceral adipose tissue, RCT: randomized controlled trial, FU: follow-up, CMR: cardiac magnetic resonance, CT: (cardiac) computed tomography.

**Table 2 nutrients-14-02926-t002:** Epicardial adipose tissue evaluation in metabolic diseases.

Study	Design	N. Pts	Population	Imaging Method	EAT Evaluation
Sato et al. [68]	RCT (FU 6 months)	35	Diabetic patients	CT	Greater ↓ EAT volume during dapaglifozin treatment than conventional therapy
Iacobellis et al. [69]	RCT (FU 6 months)	85	Diabetic patients	Echocardiography	Greater ↓ EAT during liraglutide plus metformin treatment than only metformin
Christensen et al. [70]	RCT (FU 12 weeks)	39	Patients with abdominal obesity	CMR	Both endurance and resistance training reduced EAT mass
Fernandez-del-Valle et al. [71]	RCT (FU 5 weeks)	11	Young females with obesity	CMR	Short-term, high-intensity and moderate-volume resistance training reduced EAT
Rosety et al. [72]	Prospective (FU 12 weeks)	48	Obese aged women	Echocardiography	Resistance training reduced EAT thickness
Iacobellis et al. [73]	Prospective (FU 6 months)	20	Severely obese patients	Echocardiography	Significant weight loss could be associated with a reduction in the EAT thickness, involving cardiac morphological and functional changes
Serrano-Ferrer et al. [74]	Prospective (FU 6 months)	131	Metabolic syndrome patients vs. healthy controls	Echocardiography	EAT decreased following lifestyle intervention (partly explaining myocardial function improvements)
Jo et al. [75]	Prospective (FU 8 weeks)	34	Hypertensive metabolic syndrome patients	Echocardiography	Greater ↓ EAT with high-intensity interval training than moderate-intensity continuous training
Fornieles Gonzalez et al. [76]	Prospective (FU 16 weeks)	60	Menopausal women with metabolic syndrome	Echocardiography	EAT decreased with a supervised home-based 16-week treadmill training program
Mohar et al. [77]	Cross-sectional	39	Diabetic patients	CT	Increased EAT volume was associated with the presence of severe CAD
Groves et al. [78]	Cross-sectional	362	Diabetic patients vs. healthy controls	CT	Increased EAT volume was associated with greater severity of CAD in patients with and without diabetes
Hiruma et al. [79]	RCT (FU 12 weeks)	42	Diabetic patients	CMR	Empaglifozin had similar effects as sitagliptin on EAT accumulation
Leroux-Stewart et al. [80]	RCT (FU 16 weeks)	73	Diabetic patients	Echocardiography	Greater ↓ EAT thickness during caloric restriction diet associated with physical activity
Snel et al. [81]	Prospective (FU 14 months)	14	Diabetic obese patients	CMR	EAT decreased after a 16-week low-calorie diet (reduction maintained also after 14 months on a regular diet)
Bouchi et al. [82]	Prospective (FU 12 weeks)	19	Diabetic overweight/obese patients	CMR	Luseoglifozin may impact cardiovascular risk partly by reducing the EAT volume
Iacobellis et al. [83]	RCT (FU 24 weeks)	84	Diabetic overweight/obese patients	Echocardiography	Dapagliflozin caused EAT reduction
Dutour et al. [84]	RCT (FU 26 weeks)	38	Diabetic obese patients	CMR	Exenatide caused EAT reduction
Morano et al. [85]	Prospective (FU 3 months)	25	Diabetic patients	Echocardiography	A short course of GLP-1 RA treatment induced a redistribution of EAT deposits
Elisha et al. [86]	RCT (FU 6 months)	56	Diabetic patients	Echocardiography	Greater ↓ EAT thickness with insulin detemir than insulin glargine
Murai et al. [87]	Cross-sectional	208	Diabetic patients	CT	A close relationship existed between EAT accumulation and cystatin C level
Bayomy et al. [88]	Cross-sectional	51	Diabetic patients	CMR	PAI-1 levels positively correlated with EAT volume

EAT: epicardial adipose tissue, CAD: coronary artery disease, GLP-1 RA: glucagon-like peptide-1 receptor agonist, PAI-1: plasminogen activator inhibitor-1, PTS: patients, RCT: randomized controlled trial, FU: follow-up, CMR: cardiac magnetic resonance, CT: (cardiac) computed tomography.

**Table 3 nutrients-14-02926-t003:** Epicardial adipose tissue evaluation in some specific clinical conditions.

Study	Design	N. Pts	Population	Imaging Method	EAT Evaluation
Nakanishi et al. [89]	Cross-sectional	275	CKD patients vs. non-CDK patients	CT	Greater EAT volume and high-risk coronary plaques in CKD patients
Yazbek et al. [90]	Prospective (FU 1 year)	98	Kidney transplant patients	CT	No relationship between the presence/progression of coronary calcification and EAT
Altun et al. [91]	Cross-sectional	102	Hemodialysis patients vs. healthy controls	Echocardiography	EAT thickness may be a useful indicator of early atherosclerosis
Ko et al. [92]	Prospective (FU 18 months)	109	Hemodialysis patients	CT	Lower EAT progression with Sevelamer than another calcium-based phosphate binder
Cetin et al. [93]	Prospective (FU 24 weeks)	162	Obstructive sleep apnea patients	Echocardiography	Greater EAT with AHI > 15. CPAP therapy may induce EAT regression
El Khoudary et al. [94]	RCT (FU 48 months)	474	Menopausal women	CT	Oral conjugated equine estrogens may slow EAT accumulation
Kahl et al. [95]	RCT (FU 6 weeks)	30	Depressed patients	CMR	Exercise training decreased the amount of visceral fat, in particular, EAT
Pacifico et al. [96]	RCT (FU 6 months)	51	Overweight children with NAFLD	Echocardiography	Docosahexaenoic acid supplementation decreased EAT
Farghaly et al. [97]	Cross-sectional	32	Subclinical hypothyroidism children vs. healthy children	Echocardiography	Greater EAT in children with subclinical hypothyroidism
Celik et al. [98]	Cross-sectional	75	Children with premature adrenarche vs. healthy children	Echocardiography	Greater EAT in children with premature adrenarche (positively correlated with DHEA-SO4 level)
Longenecker et al. [99]	Cross-sectional	118	HIV patients	CT	EAT volume and density were related to insulin resistance at baseline

EAT: epicardial adipose tissue, CKD: chronic kidney disease, AHI: apnoea-hypopnoea index, CPAP: continuous positive airway pressure, NAFLD: non-alcoholic fatty liver disease, DHEA-SO4: dehydroepiandrosterone sulfate, PTS: patients, RCT: randomized controlled trial, FU: follow-up, CMR: cardiac magnetic resonance, CT: (cardiac) computed tomography.

## Data Availability

Not applicable.

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
