# Peer review of "Epicardial Adipose Tissue: A Novel Potential Imaging Marker of Comorbidities Caused by Chronic Inflammation"

_nutrients, 2022, doi:10.3390/nu14142926_

Round 1

Reviewer 1 Report

The article is well structured and written and expected to attract broad readership. There is considerable increase of current literature regarding the discussed topic and the authors have comprehensively covered most references. I have the following very minor comments that can be considered to further improve the outcomes:

1) Figure 1 looks incomplete. Please consider furnishing with mechanistic endpoints rather than a singularity linear reaction.

2) Kindly mention source of Fig 4.

3) General organization of a review article differs from a research article in a sense, a 'material and methods' section is not required. Please consider restructuring the article. 

Author Response

Dear Reviewer,

thank you very much for your comments and suggestions that allow us to improve the MS.

In order to answer to you comments we modified the figure 1, include the source of the figure 4.

in relation to the structure of the Review, even if it is not necessary the Matherial and Methods section, in our opinion, it is more clear for a systematic reading of the MS.

Thank you and kind regards

Reviewer 2 Report

This paper is weel organice and it suold be of interest for medila practise